# Dipole and Convergent Single-Well Thermal Tracer Tests for Characterizing the Effect of Flow Configuration on Thermal Recovery

**Jérôme de La Bernardie** [1,*] , **Olivier Bour** [1] , **Nicolas Guihéneuf** [1,2] , **Eliot Chatton** [1,3] , **Laurent Longuevergne** [1] **and Tanguy Le Borgne** [1]

1. Univ Rennes, CNRS, Géosciences Rennes-UMR 6118, 35000 Rennes, France; olivier.bour@univ-rennes1.fr (O.B.); guiheneuf.nicolas@gmail.com (N.G.); eliot.chatton@gmail.com (E.C.); laurent.longuevergne@univ-rennes1.fr (L.L.); tanguy.le-borgne@univ-rennes1.fr (T.L.B.)
2. G360 Institute for Groundwater Research, College of Engineering and Physical Sciences, University of Guelph, Guelph, ON N1G 2W1, Canada
3. Sorbonne Universités, UPMC Univ Paris 06, UMR7621, Laboratoire d'Océanographie Microbienne, Observatoire Océanologique, 66650 Banyuls/mer, France
* Correspondence: jerome.la.bernardie@gmail.com

**Abstract:** Experimental characterization of thermal transport in fractured media through thermal tracer tests is crucial for environmental and industrial applications such as the prediction of geothermal system efficiency. However, such experiments have been poorly achieved in fractured rock due to the low permeability and complexity of these media. We have thus little knowledge about the effect of flow configuration on thermal recovery during thermal tracer tests in such systems. We present here the experimental set up and results of several single-well thermal tracer tests for different flow configurations, from fully convergent to perfect dipole, achieved in a fractured crystalline rock aquifer at the experimental site of Plœmeur (H+ observatory network). The monitoring of temperature using Fiber-Optic Distributed Temperature Sensing (FO-DTS) associated with appropriate data processing allowed to properly highlight the heat inflow in the borehole and to estimate temperature breakthroughs for the different tests. Results show that thermal recovery is mainly controlled by advection processes in convergent flow configuration while in perfect dipole flow field, thermal exchanges with the rock matrix are more important, inducing lower thermal recovery.

**Keywords:** Single-well thermal tracer test; geothermal energy; Fiber-Optic Distributed Temperature Sensing; Plœmeur site

---

## 1. Introduction

Heat has been widely used as active and passive tracers for the characterization of thermal transport and hydrological processes in aquifers [1–7]. For instance, it has been shown in many examples during the last 20 years that the use of heat is very efficient for characterizing surface-groundwater exchanges [6,8]. The use of heat as an active tracer has been also developed for geothermal purpose. In particular, long-term injection tests in porous media were carrying out to characterize the efficiency of geothermal reservoir for thermal storage such as Aquifer Thermal Energy Storage (ATES) [9,10]. Heat tracer tests appeared also very informative to determine hydrodynamic properties and characterize aquifer heterogeneity [1,2,7,11]. One of the main advantages of the use of heat as a groundwater tracer

in the sub-surface is the possibility to image spatially and temporally the spread of a thermal plume thanks to geophysical methods, in an aquifer with Electrical Resistivity Tomography (ERT) [2,12] and at borehole scale, with Fiber Optic-Distributed Temperature Sensing (FO-DTS) [13], with very good spatial and temporal resolutions.

Heat has been also used to characterize heterogeneities in complex aquifers like karstic or fractured media. Borehole temperature profiles often show temperature changes caused by water of warmer or colder origins flowing from permeable flow zones [14–18]. Groundwater flow occurring in a permeable fracture may lead to a temperature anomaly in the temperature profile that is a function of groundwater velocity [18]. Alternatively, in the case of several inflows and outflows within the boreholes, ambient flow in the borehole can also greatly modify temperature logs [16,19]. Some other studies have shown that borehole temperature profiles under induced fluid flow conditions by pumping in an adjacent borehole could be used to estimate fracture interconnections [20] and hydraulic properties [21]. Very recently, thermal tracer tests have also shown their efficiency to demonstrate the impact of flow channeling on heat transport [5,22,23]. In particular, it has been shown that flow channeling may highly reduce thermal transit times and improve thermal recovery during thermal tracer tests [5,22]. Nevertheless, it remains very challenging to achieve thermal tracer tests in fractured crystalline media due to both the relatively low permeability of fractured rocks in general [24–26] and the important thermal lag time and attenuation induced by conduction of heat within the matrix [5,21,27]. As a consequence, although heat tracer tests in fractured media are crucial for predicting and optimizing geothermal storage and extraction in crystalline rock, only few thermal tracer tests have been achieved in fractured media to our knowledge, and little is known about thermal injection and recovery in such complex media. In particular, the role of flow configurations on thermal recovery during thermal tracer tests has not been investigated experimentally.

Two main flow configurations may be used during thermal tracer tests: perfect dipole, where the injection flow rate equals the withdrawal flow rate [5,22], or convergent dipole, where the injection flow rate is significantly lower than the withdrawal flow rate [28]. Both configurations are generally used for different applications. For instance, the convergent flow field configuration has been widely used during solute tracer tests for characterizing transport properties [29–31] and to insure high recovery rates. On the other hand, the perfect dipole configuration is usually applied in geothermal doublet [32] and Enhanced Geothermal System (EGS) [33] for heat storage or extraction. However, it has been recently shown that transport in fractured media is highly influenced by flow conditions. In particular, Kang et al. [30] demonstrate that flow distribution controls the late time scaling of breakthrough curves. The flow distribution also highly controls the exchange surface between the fractures and the matrix, and thus fracture/matrix thermal exchanges should be very different in function of the flow conditions. Thus, both flow configurations (perfect dipole and convergent dipole) should lead to different thermal recoveries and may have different interests for the characterization of thermal transport. On the one hand, convergent thermal tracer tests may enhance streamline convergence to the outlet. On the other hand, perfect dipole flow tests spread heat over a larger volume of rock, enhancing heat exchanges with the rock matrix. In both cases, energy injected should be partly restituted during the pumping phase but, the timing of energy recovery may be quite different depending on the flow configuration.

Thus, the objective of our study is to analyze the impact of flow configuration on thermal recovery during thermal tracer tests, which may be very helpful to choose the appropriate flow configuration in function of the targeted application and the media properties. In the following, we present several thermal tracer tests with continuous injection achieved in the same borehole in various flow conditions: (i) Perfect dipole test and (ii) Convergent dipole test. Note that the results of these tests were already partly presented in de La Bernardie et al. [5] to compare solute tracer tests with thermal tracer tests and analyze the influence of fracture geometry on thermal transport. Here, the present study aims at

characterizing the impact of flow on thermal transport. In the following, we first describe the field site and the methodology, then the results and the interpretation of thermal recovery are presented. We also highlight the interest of using FO-DTS to achieve and interpret thermal tracer tests.

## 2. Methodology

Thermal tracer tests are usually achieved in cross-borehole configuration [2,22,28]. However, as shown by de La Bernardie et al. [5], single-well thermal tracer test can be an alternative to cross-borehole tracer test. It consists of injecting hot water within a fracture isolated by a straddle packer (or double packers) and withdrawing water above in the same borehole. Such vertical dipole configuration has already been developed in porous media for hydraulic and solute tracer tests to characterize the vertical hydraulic heterogeneity of porous aquifer [34–36]. In the present study, we choose this experimental setup since the excellent hydraulic connection between two fractures crossing the same borehole was confirmed thanks to previous tracer tests using dissolved helium and amino G acid [37]. As we targeted the two shallowest flowing fractures of the borehole (B3-1 and B3-2), localized thanks to previous hydraulic experiments [38], we did not isolate the upper fracture from the top part of the borehole. The setup used is fully described in de La Bernardie et al. [5] and briefly explained below.

Performing injection and withdrawal along the same borehole complicates the interpretation and data processing because the increase of temperature due to thermal losses along the injection tube during the injection phase may disturb the heat recovery signal. Thus, before estimating thermal recovery for the different thermal tracer tests, caution should be taken for identifying the thermal breakthrough curves from the raw temperature data. A method to properly process the temperature data and extract the corrected thermal breakthrough curve is thus first presented. Then, the estimates of thermal recovery for the different flow configurations are compared after applying the correction. Fiber-Optic Distributed Temperature Sensing (FO-DTS) is also used with success to monitor temperature with high spatial and temporal resolution [39,40] and effectively localize heat arrival. In addition, this innovative instrumentation is shown to be very useful for temperature monitoring as it allows us to better separate the increase of temperature due to heat losses along the injection tube, from the thermal tracer transported within the fracture network.

### 2.1. Field Site

The single-well thermal tracer tests were carried out at Stang er Brune, which is part of the Plœmeur field site belonging to the French hydrogeological observatories network H+ [41] and the French Network of Critical Zone Observatories OZCAR (Observatoires de la Zone Critique: Applications et Recherche) [42] The Plœmeur site corresponds to a fractured crystalline aquifer that provides ~ 1 million $m^3$ per year to supply drinking water to the city of Plœmeur (20,000 habitants) [43] (Figure 1a). The experimental site of Stang er Brune, which is not influenced by the pumping, is located near the outcrop of a permeable contact zone between granite and mica-schist which deeps toward north. Three open boreholes of about 100 meter deep, cross the contact zone at 40 meters deep (Figure 1b). The high connectivity and transmissivity of fractures at different depths allowed the implementation of innovative experiments [23,30,38,44] during the last few decades, to characterize the impact of heterogeneity of fracture media properties on flow and transport.

The heat tracer experiments were carried out in the B3 borehole between B3-1 and B3-2 fracture. B3-1 is a large fracture zone between 37.5 and 33.6 meter deep with a transmissivity of $7.0 \times 10^{-4}$ $m^2$/s while B3-2 is located at 44.5 meter deep with a transmissivity of $2.4 \times 10^{-3}$ $m^2$/s [5,37,38] (Figure 1c). The good fracture transmissivities and connectivity allow the circulation of fluid between the fractures at a large range of flow rates.

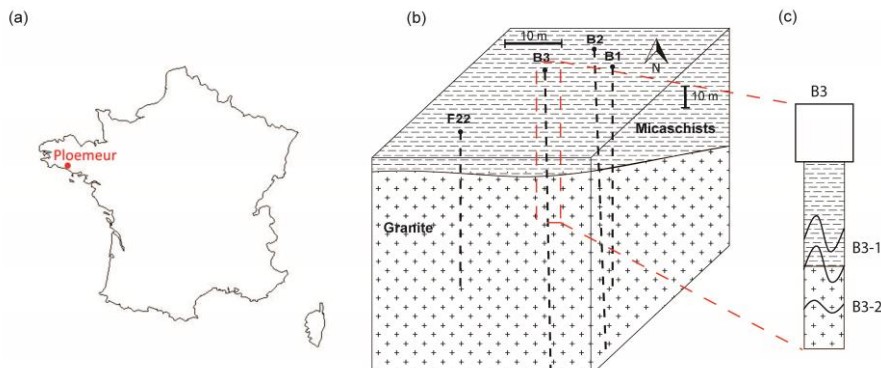

**Figure 1.** (**a**) Localization of the Ploemeur field site, (**b**) Geological diagram of Stang er Brune experimental site with the locations of the different boreholes and (**c**) geological log with main fracture localization derived from optical and acoustic logs.

## 2.2. Experimental Setup

A straddle packer (here two packers) installed in B3 was used to hydraulically isolate the B3-2 fracture from B3-1 and from the bottom part of the borehole. The water was pumped above the straddle packer and injected in a tank of 200 liters for the perfect dipole test, and of 3 m3 for the convergent dipole tests. The water in the tank was then pumped to a mobile water flow heater (Swingtec AquaMobil DH7), which heated the fluid up to 60 °C, before the injection in the chamber of the straddle packer (Figure 2). This set up allowed the continuous circulation of the fluid in the media during the whole experiment. A rubber sleeve was attached on the injection tube to reduce heat losses in the water column. Insulate tape was used at the junctions of the sleeves. During the tests, the temperature was monitored using the Fiber-Optic Distributed Temperature Sensing (FO-DTS) installed in the upper part of the borehole from the surface to the top of the straddle packer positioned at 43 m deep (Figure 2). An Ultima S (SILIXA manufacturer) DTS unit was used to monitor temperature, with a spatial sampling of 12.5 cm and a time integration of one minute. Additional RBR T temperature sensors (0.002 °C accuracy) were used to calibrate the raw data into temperature using the post processing single ended calibration procedure [45]. The experimental set up has been previously presented in detail in de La Bernardie et al. [5].

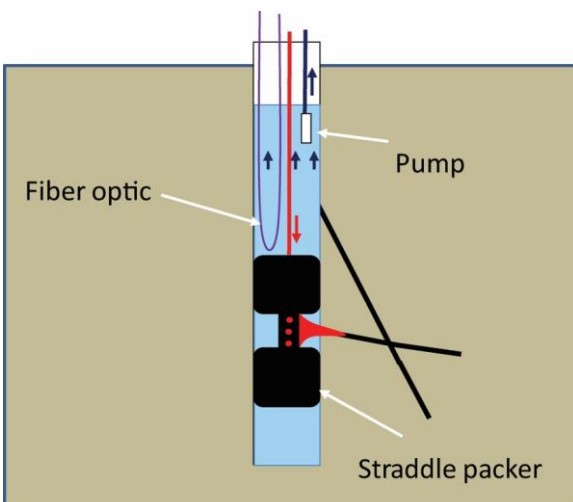

**Figure 2.** Experimental set up of the single-well thermal tracer test.

Whereas in de La Bernardie et al. [5], focus was on a pulse heat tracer test experiment and on the comparison of heat and solute transport, here, three additional thermal tracer tests with continuous injection in a single-well configuration are presented. Two tests (Experiments I and II, Table 1) were

achieved in a convergent dipole flow field at two different pumping flow rates (125 l/min and 90 l/min) and one test was performed in a perfect dipole flow field at a flow rate of about 15 l/min (Experiment III, Table 1). For each configuration, the injection flow rate was between 15 to 16 l/min. Heated water was injected continuously between 2 and 5 hours. Note that for Experiment III, injection flow rate (16 l/min) was slightly different from the pumping flow rate (22 l/min), but we assume here that the flow conditions do not differ significantly from a perfect dipole flow field.

After the end of the injection, the pump was running throughout the night to recover most of heat that have been injected and to obtain thermal breakthrough records as complete as possible. Table 1 presents the experiment parameters of the different tests. For each experiment, pumping and injection of water at ambient temperature were started for at least one hour preceding the injection of heat, to insure hydraulic quasi-steady state during thermal tracer tests.

As mentioned before, raw FO-DTS temperature measurements during the experiments can be disturbed by heat losses all along injection tube, especially because the fiber optic is directly taped against the sleeves isolating the injection tube. To better localize and estimate thermal losses in the water column, an injection test was carried out. This test consists of injecting heated water during two hours without pumping above the double packer (Experiment IV in Table 1). During this experiment, the increase of temperature in the upper part of the borehole (i.e., above the straddle packer) was thus only due to heat losses along the injection tube since no pumping occurred. As we shall see in Section 3.2, this experiment was essential to interpret thermal tracer tests with continuous input as it allowed us to properly distinguish temperature rises due to advective signal (thermal breakthrough) from heat losses along the injection tube. All the experiments were done during a two-week field campaign.

### 2.3. Conceptual Scheme of the Flow Configurations

Convergent and perfect dipole configurations are associated to different streamline geometries which depend on boundary conditions (injection and withdrawal flow rates) and media properties. As here, we analyze the same media under different boundary conditions, the streamline shape should differs only due to injection and withdrawal flowrates. As transport of heat is partly controlled by streamline geometry, both flow configurations are expected to lead to different thermal recoveries.

To better highlight the effect of flow configurations on streamlines and thus on thermal recovery, Figure 3 shows a conceptual scheme of the experiments in convergent and perfect dipole configurations. For simplicity, the scheme of Figure 3 represents streamlines in homogeneous media. In fractured media, streamlines may be much more complex and dependent on local fracture geometry. For the convergent dipole tracer tests, withdrawal flow rates were higher than injection flow rates (I and II in Table 1) while for the perfect dipole tracer test (III in Table 1), injection and withdrawal flow rates were similar. Because the amount of water withdrawn equals the amount of water injected for the perfect dipole test, all water injected should be withdrawn and the streamlines are expected to have spherical shapes [46,47] (Figure 3a). In this configuration, recovery of a conservative solute tracer should be perfect while heat recovery should only depend on heat conduction along the streamlines. For the convergent case, where the pumping rate is higher than injection rate, the pumping withdraws water from a larger volume of investigation and not only from the injection chamber [29] (Figure 3b). In this configuration, temperature is expected to be diluted proportionally to the pumping flow rate while the spreading of heat will be limited to few streamlines around the borehole (Figure 3b). Thermal conduction in the surrounding rock will only take place in the vicinity of the well, along the streamlines that connect the injection to the withdrawal chamber as shown in Figure 3b. To summarize, in perfect dipole configuration, heat recovery may be more affected by the spreading of heat and conduction within the rock matrix, while in convergent dipole thermal tracer test, breakthrough curve may be principally controlled by advection. Note that in the present study, flow conditions are comparable to the conceptual scheme of Figure 3 only during the heat injection. During the recovery period, only pumping occurs, which should change streamline shape and flow pathways. In convergent dipole, streamlines should not change significantly as flow field is weakly disturbed by injection due to the

predomination of pumping flow rate. In perfect dipole, when injection is stopped, the streamlines which spread heat in a large volume away from the borehole are not in function anymore and only heat stored in the neighborhood of the well should be recovered.

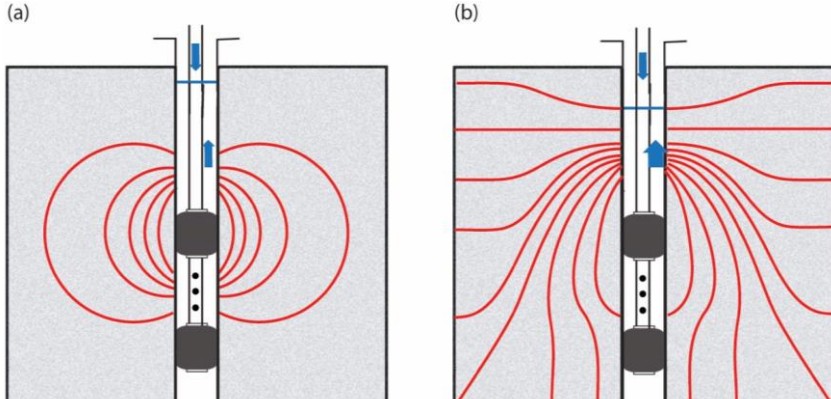

**Figure 3.** Conceptual scheme and streamlines during single-well flow tests in (**a**) perfect dipole configuration and (**b**) convergent dipole configuration.

**Table 1.** Experimental set up of the three heat tracer tests and the heat injection test.

|  | I | II | III | IV |
|---|---|---|---|---|
| **Flow configuration** | Convergent | Convergent | Perfect dipole | Divergent |
| **Type of heat injection** | Continuous injection | Continuous injection | Continuous injection | Continuous injection |
| **Heated water injection rate** | 16.2 l/min | 16.7 l/min | 16 l/min | 15.1 l/min |
| **Pumping rate** | 125 l/min | 90 l/min | 22 l/min | |
| **Injection temperature** | 57.1 °C | 55.9 °C | 56.7 °C | 50.9 °C |
| **Duration of heat injection** | 4 h 50 min | 2 h 10 min | 4 h 02 min | 2 h 02 min |
| **Duration of heat recovery** | 15 h 14 min | 13 h 10 min | 17 h 24 min | |

### 2.4. Calibration and Data Processing

For calibrating the FO-DTS, four coiled sections of the cable were plunged, during the whole duration of the tests, in two calibration baths filled with water at ambient temperature (15 °C) and wetted ice (0 °C) [5]. RBR solo T temperature sensors (0.002 °C accuracy) allowed us to monitor the temperature in each bath for FO-DTS calibration. Three reference coiled sections situated before the splicing of the fiber optic were used for the calibration, two in the cold bath and one in the ambient bath. The RBR solo temperature probes allowed to convert the laser backscattered signal measured by the FO-DTS to temperature using the single ended calibration procedure [45], with a very satisfactory temperature resolution (temporal mean of temperature variability along the cable) in the calibration bath (0.08 °C). The FO-DTS calibration processing is the same as the one used for the heat pulse single-well thermal tracer test described in de La Bernardie et al. [5].

The main challenge here is to differentiate the temperature breakthrough curve resulting from advective transport through the fracture network from the increase of temperature due to heat losses along the injection tube. To better highlight these distinct processes during the experiments and complete information provided by the injection test (Experiment IV in Table 1), we applied Karthunen–Loève Transform (KLT) [48] to the temperature data.

Data processing using Karthunen–Loève Transform (KLT) allows us to extract uncorrelated characteristic temporal signals from space-time data field. It consists in defining a new orthogonal basis where each eigenvector is representative of part of the data variance. In this study, the data field corresponds to the evolution with time of the measured temperature in the borehole. Temperature is not an extensive variable, which is a prerogative for applying KLT to filter data. Therefore, KLT will

only allow to qualitatively verify the first hypothesis about the presence of both signals and determine the global ratio between diffusive signal due to heat losses and advective signal from the thermal breakthrough curve.

To quantitatively remove the temperature signal due to thermal losses along the tube from the raw FO-DTS measurements, we used a very simple method. During a single-well thermal tracer test with continuous injection, the temperature signal visible at first is the heat losses along the injection tube before the arrival of the thermal breakthrough. We supposed that the temperature of those heat losses becomes rapidly constant as advection processes along the tube cools this conductive signal and limits transversal conduction in the borehole [49]. The signal measured before thermal breakthrough arrival can thus be considered as a "temperature background" that does not vary with time. Then, we directly subtract this constant background temperature to the measured temperature to extract the thermal breakthrough of the tracer tests.

## 3. Results

### 3.1. Temperature Breakthrough of Experiment III

We describe here in detail only the results of Experiment III, but this section is also useful to understand the thermal records of Experiments I and II. Figure 4b presents the raw temperature records acquired by FO-DTS during the perfect dipole test (Experiment III in Table 1). The x-axis represents the time during the experiment while y-axis corresponds to the depth in the borehole. To simplify the reading, injection starts at time t = 0. The temperature in the injection zone was at ambient water temperature before and after the injection, and reached 57 °C during heat injection from t = 0 to t = 4 hours (Figure 4c). Note that during the recovery period, we injected water at ambient temperature for 1 hour, 6 hours after the start of the test (Figure 4c). Three main phases are then observed above the injection chamber (Figure 4b): before heat injection, during heat injection and during the thermal tracer recovery when only pumping occurs.

Before heat injection, background temperature is at about 16 °C. The pumping in the borehole induces mixing with a constant temperature all along the borehole. Then, after the injection starts, the temperature increases in all the water column. An important and relatively regular increase in temperature occurred along the borehole above 34.5 meters, from 16 °C to about 23 °C. Several hot stripes can be observed all along the borehole with a hotter zone below 37.5 m. Between 34.5 and 37.5 m, one may also observe a colder zone with the presence of few stripes. These stripes mainly come from thermal losses along the injection tube, which heats the fiber optic by conduction. Despite the presence of insulate tape, the stripes appear mainly located at the junctions between the sleeves, which insulated the injection tube during all experiments. The greater temperature variations below 37.5 m take place in front of the steel tube of the packer, which was not insulated with the sleeves and consequently has a greater thermal conductivity.

After the end of the injection, temperature decreases at different velocity depending on the zone in the borehole. Between 33.6 and 34.5 m, a hot spot where temperature remains relatively high can be observed, even after the end of the injection. In the next section, it will be demonstrated that this signal is the thermal recovery of the thermal tracer injected at B3-2. Before the end of the injection, this hot spot cannot be easily identified due to the superposition of the stripes, which also increase the temperature measured by the FO-DTS. The temperature signal at this depth, resulting from the thermal breakthrough curve superimposed with the stripe signal, is displayed in Figure 4d. In the water column, above 33.6 m, temperature decreases directly after the end of injection. Between 34.5 and 37.5 m, water temperature reaches more rapidly the ambient temperature as soon as heat injection stops. As illustrated below, the low temperature in this area during and after the injection can be explained by the presence of an ambient water inflow that decreases the temperature by dilution. This rapid water cooling in this area shows also that heat transfer from B3-2 to this part of B3-1 is negligible for the duration of the observation. At depth greater than 37.5 m, temperature decreases

slowly from 24 °C to the ambient temperature, suggesting the presence of an immobile zone where heat is transported in water and rock only by conduction.

Those observations are consistent with the presence of the main inflow zones between 33.6 and 37.5 m, just above the no flow area below 37.5 meters. These two main inflows are characterized by an inflow at ambient temperature between 34.5 and 37.5 meters (B3-1B) and a hot inflow between 33.6 m and 34.5 meters (B3-1A) [5]. The hot inflow B3-1A corresponds to the fast advection of heat from B3-2 fracture and can be associated to the thermal tracer breakthrough. The cold inflow B3-2B may correspond to another pathway either connected to B3-2 with high thermal attenuation or disconnected from B3-2. Above the B3-1 fracture, cold and heat inflows are mixed and advected in the borehole due to the constant pumping.

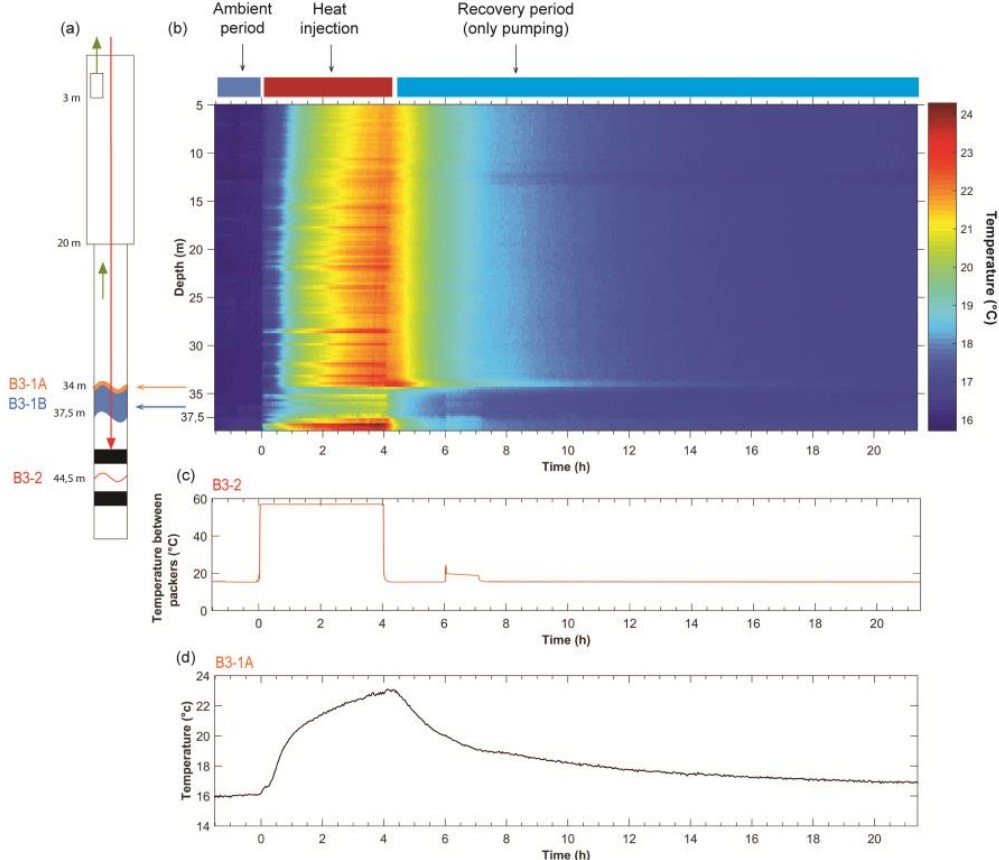

**Figure 4.** (**a**) B3 borehole scheme with the position of the pump (3 m), the injection fracture B3-2 (in red), the hot inflow B3-1A (in orange) and the cold inflow B3-1B (in blue) interpreted from (**b**) the measured temperature with the fiber-optic distributed temperature sensing (FO-DTS) during Experiment III, where the *x* axis represents the time during the experiment, and the *y* axis corresponds to the depth in the borehole, (**c**) temperature measured in the injection chamber at B3-2 depth and (**d**) thermal breakthrough superimposed with thermal losses signal measured with the FO-DTS at B3-1A (34 m).

*3.2. Characterization of Heat Losses from the Injection Tube During the Heat Injection Test*

To characterize and localize heat losses that occur during heat injection all along the injection tube, we achieved a heat injection test without pumping (Experiment IV in Table 1). Figure 5a displays the temperature records measured by FO-DTS along the injection tube during Experiment IV. Rapidly after injection start, stripes of variable temperatures appear due to discontinuities in injection sleeve insulation as mentioned previously in Section 3.1. Figure 5b presents three temperature profiles measured by FO-DTS: In ambient condition before the beginning of the injection, 1 h 30 min after heat injection starts without any pumping and 1 h 30 min after heat injection starts with pumping during

Experiment III. This demonstrates that after the same injection duration, a difference of more than two degrees can be observed between both tests in the water column above 34 m. This difference confirms the presence of a hot inflow coming from the fracture at 34 m (B3-1A fracture). At B3-1B depth, an increase of temperature is also highlighted between both tests. This observation confirms that B3-1B is connected to B3-2, but with higher thermal attenuation.

The heat injection test was thus essential to properly identify the effect of heat losses signal on the temperature records during the thermal tracer tests. The comparison with another experiment involving pumping confirms that the increase of temperature is the superposition of two signals: the temperature increase due to thermal losses along the injection tube and the thermal breakthrough. Nevertheless, the temperature recorded during the heat injection test cannot be directly used to determine the fraction of heat that comes from each process. During this test, no flow occurred along the injection tube while during the thermal tracer tests, the flow in the borehole also impacts the thermal losses signal measured by the FO-FTS. In pumping configuration, the increase of temperature due to thermal losses is expected to be lower than without pumping due to advection of colder water along the tube. In the following, a method is proposed to distinguish and separate thermal tracer breakthrough from heat losses signal along injection tube.

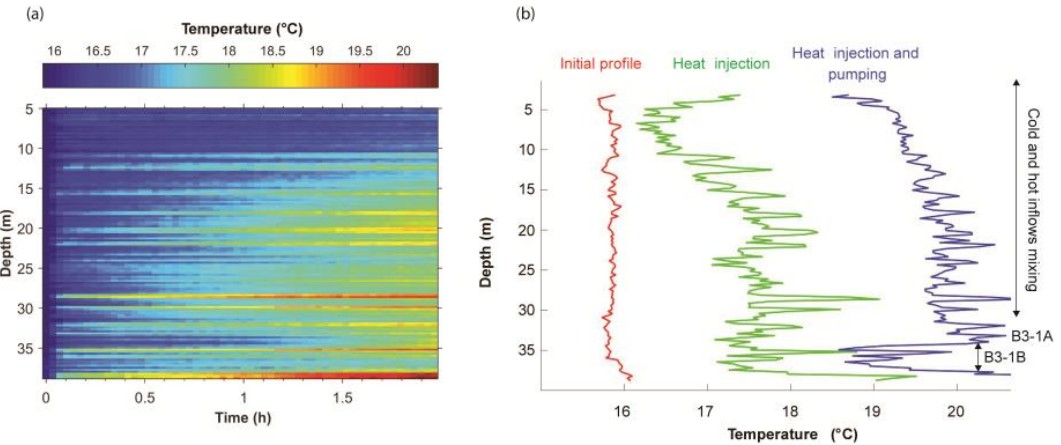

**Figure 5.** (**a**) Evolution of the temperature measured by FO-DTS in the borehole during Experiment IV and (**b**) different temperature profiles measured by FO-DTS in the borehole after 1 h 30 min of heat injection during Experiment IV (green) and during Experiment III (blue), compared to the initial profile (red).

### 3.3. Data Processing Using Karthunen–Loève Transform

Karthunen–Loève Transform [48] was applied to process the data of Experiment III in order to differentiate thermal tracer breakthrough (advective signal) from the heat losses along the injection tube. Using this data processing, two main eigenvectors can explain the signal as illustrated in Figure 6. These signals have two different signatures. The first eigenvector, which participates to 95% of the variance of the raw FO-DTS measurements, displays a slow increase in the amplitude. The shape of this eigenvector may be thus associated to the thermal breakthrough. The second eigenvector represents a step function that begins at injection start and finishes at injection stop. The shape of this eigenvector is typical of a conductive heat flux perpendicular to a flowing fluid [49]. The second eigenvector certainly represents the heat losses coming from the injection tube. This data processing demonstrates the presence of two distinct signals that participate to the total temperature measurements. In addition, the amount of variance explained by the each eigenvector indicates that the temperature breakthrough coming from the fracture clearly predominates the total signal compared to heat losses coming from the injection tube. Note that the KLT method is a purely mathematical decomposition and the amplitude of the eigenvalues (Figure 6) does not have direct physical meaning.

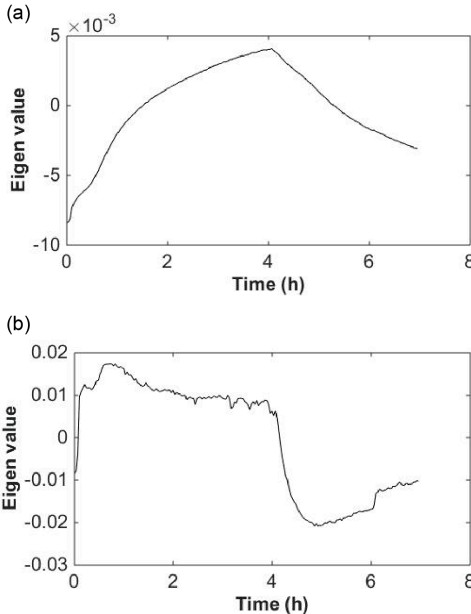

**Figure 6.** Data processing using Karthunen–Loève Transform on Experiment III results: (**a**) first and (**b**) second eigenvector, which can be associated to thermal tracer breakthrough signal and heat losses signal along injection tube, respectively.

As explained in Section 2.3, the KLT method does not allow to quantitatively separate thermal tracer breakthrough from heat losses signal along injection tube. To remove the increase of temperature due to heat losses, from the thermal breakthrough curves of Experiments I, II and III, we simply subtract the early constant temperature recorded to the raw FO-DTS measurements. Figure 7 displays the raw and the corrected temperature anomalies (difference between measured temperature and initial temperature) during Experiment III. At 34 m, the mean temperature increase due to heat losses is 0.46 °C which relatively low compared to the one coming from the thermal breakthrough. For the case of Experiment III, by simple energy balance calculation, the mean thermal losses signal along the tube of injection correspond to about 6% of the total heat recovery, which confirms the result of the data processing using Karthunen–Loève Transform. Although Experiment III is the only one described in detail, similar results were obtained for Experiments I and II. Note however that, in Tests I and II, the temperature increase due to heat losses will be lower than for the case of Test III, due to the higher flowrates which dilute the signal.

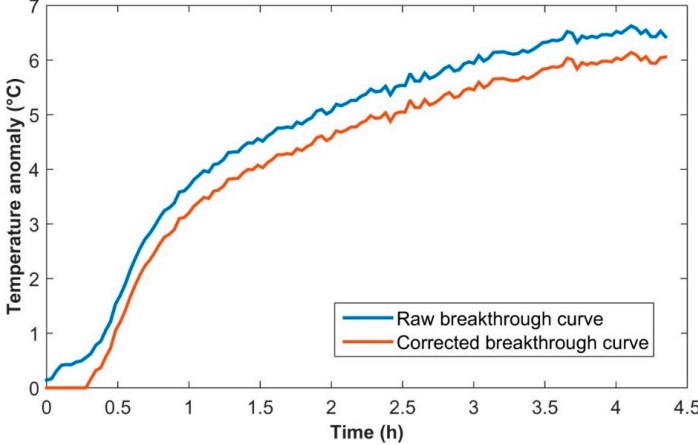

**Figure 7.** Thermal breakthrough curves measured at B3-1A depth during Experiment III before (blue) and after (red) removing the signal of heat losses coming from the injection tube.

### 3.4. Comparison of the Thermal Breakthroughs

To illustrate the influence of flow rate and flow configuration on thermal recovery, the raw FO-DTS measurements during Experiment I, II and III are compared using the same color scale in Figure 8. This figure shows that temperature signal is more attenuated when the flow rate increases, which is simply due to an increase of thermal dilution with the flow rate. For each experiment, the high temperature signal from thermal tracer breakthrough can be localized at 34 m. This heat inflow is much more visible for high flow rates (Figure 8a,b) as thermal losses signal along injection tube have less influence on temperature records due to thermal dilution.

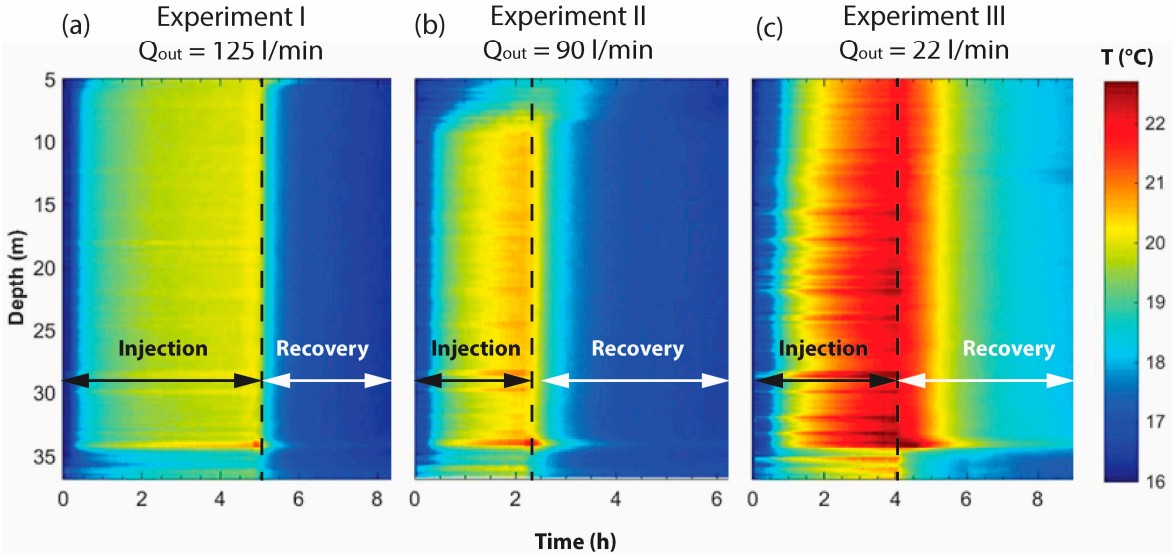

**Figure 8.** Raw FO-DTS measurements during and after heat injection for (**a**) convergent dipole configuration at Q = 125 l/min, (**b**) convergent dipole configuration at Q = 90 l/min and (**c**) perfect dipole configuration at Q = 22 l/min.

To compare the thermal recovery during each thermal tracer test, the thermal recovery rate has been calculated for each experiment. The thermal recovery rate, is the ratio between instantaneous recovery power, *P* in W and injection power $P_{in}$ in W, calculated as:

$$\frac{P(t)}{P_{in}} = \frac{c_w \rho_w \times Q_{out} \times (T(t) - T_0)}{c_w \rho_w \times Q_{in} \times (T_{in} - T_0)} = \frac{Q_{out} \times (T(t) - T_0)}{Q_{in} \times (T_{in} - T_0)}, \tag{1}$$

where $c_w$ and $\rho_w$ are respectively the specific heat and density of water, $Q_{out}$ is the withdrawal flow rate in $m^3/s$, $Q_{in}$ is the injection flow rate in $m^3/s$, T is the corrected temperature breakthrough at B3-1A in °C, $T_0$ is the initial temperature and $T_{in}$ is the temperature of injection in °C. Figure 9 shows the evolution with time of the transfer functions for the different flow configurations. From this calculation, heat recovery during the injection phase appears to be higher for convergent cases, with 58% of thermal energy recovery for highest flow rate and 15% for perfect dipole test. After injection stops, heat recovery declines very rapidly for convergent configurations, but in the case of perfect dipole, the recovery lasts for much longer times and the tailing slope is lower. In the convergent case, the transfer function becomes slightly negative during the recovery stage. This phenomenon is expected to be induced by the pumping, which carried water slightly colder compared to the initial background temperature. Applied energy balance suggests that differences between configurations are small during the recovery period, with a thermal energy recovery slightly higher for the convergent case with a ratio of 37% against 34% for perfect dipole configuration. However, for convergent configuration no more heat was recovered after 12 h of thermal recovery from stored energy while for the perfect dipole case, more heat could have been recovered if the pumping had lasted longer.

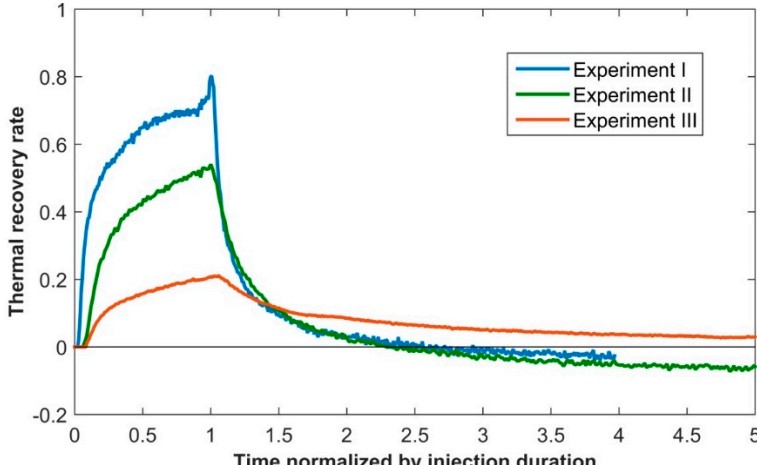

**Figure 9.** Thermal recovery rate for the different experiments as a function of the time normalized by the injection duration at B3-1A depth (34 m).

## 4. Discussion and Conclusions

This study presents temperature data from three single-well thermal tracer tests achieved with continuous injection but in different flow configurations from convergent dipole and perfect dipole flow field, in order to analyze the role of flow configuration on thermal recovery. The experimental set up is similar to the one used in de La Bernardie et al. [5], which focuses on a heat pulse tracer test to analyze the role of fracture geometry on thermal transport. Due to the long duration of the injection phase for the experiments presented here, a heat injection test and additional data processing methods were also presented to better identify the tracer thermal breakthrough from temperature data that are also slightly disturbed by heat losses along the injection tube.

Once temperature data were properly processed, the comparison of the three thermal breakthroughs illustrated that thermal recovery was more efficient in convergent configuration during the injection period despite the lower temperature signal measured by the FO-DTS. During the recovery period, the tailing slope of the breakthrough curve was, however, significantly lower in perfect dipole configuration, suggesting longer restitution of the energy stored during injection phase. These results suggest that for the convergent case, heat recovery is mainly controlled by advective processes due to the shorter transit times. However, during perfect dipole tracer test, heat exchanges by conduction within the matrix seem more significant, potentially due to i) a longer time for heat exchanges during heat transfer and ii) a possible greater spreading of the thermal tracer over a greater exchange surface.

During the recovery phase, convergent and perfect dipole tests had almost similar thermal energy recovery. We would expect, however, to recover more heat in convergent dipole because in this case transit times are lower and spreading of heat may be limited to few streamlines confined around the borehole (Figure 3). The low recovery in convergent dipole may be due to the uncertainty associated with the possible variations of the background temperature, which may limit the quantitative estimates of energy recovery (Figure 9).

To better study the role of flow configuration on energy recovery during thermal tracer tests, few more experiments performed under different injection flow rates and pumping flow rates would be very useful. In Section 2.3, the surface of exchange was assumed to be higher in the case of the perfect dipole configuration due to the shape of the streamlines (Figure 3). It would be particularly interesting to better evaluate the roles of the surface of exchange and transit times on the observed contrast between breakthrough curves. The convergent and perfect dipole tracer tests were not achieved at the same pumping flow rate as the equipment limited the injection flow rate to 15 l/min. Therefore, the effects of transit time and exchange surface on thermal recovery cannot be distinguished from the results of those thermal tracer tests. Additional perfect dipole tests achieved with larger injection and

pumping flow rates (90 l/min or 125 l/min) would have been particularly interesting to study these effects. Analytical and numerical simulations could be also very useful.

From these observations, some recommendations can be provided to help the implementation of thermal tracer tests. It is generally more appropriate to do thermal tracer tests in convergent configuration for studying thermal transport due to the lower thermal transit times which allow limited time of experiment and thermal recovery to be optimized. The important dilution of the temperature signal observed in this configuration may, however, decrease the amplitude of the signal below the detection limit of the used temperature sensor. When fracture permeability and geometry allow short transit time, especially when flow channeling occurs for example [5], the perfect dipole configuration may be ideal for studying thermal storage or extraction, as it may give more information about diffusive processes within the rock matrix and it limits temperature signal dilution. This configuration is also analogous to the one used in geothermal doublet and EGS and could be very useful for assessing the possibility of geothermal energy development.

**Author Contributions:** Conceptualization, J.d.L.B., O.B. and T.L.B.; Formal analysis, J.d.L.B. and L.L.; Funding acquisition, O.B.; Investigation, J.d.L.B., O.B., N.G., E.C. and T.L.B.; Methodology, J.d.L.B., O.B., N.G., E.C. and T.L.B.; Validation, J.d.L.B. and O.B.; Writing—original draft, J.d.L.B.; Writing—review & editing, J.d.L.B., O.B. and N.G.

**Funding:** Funding was provided by the ANR project Stock en Socle (grant ANR-13-SEED-0009), the network of hydrogeological observatories H+ (hplus.ore.fr/en/) and the ANR project EQUIPEX CRITEX (grant ANR-11-EQPX-0011).

**Acknowledgments:** The whole dataset is available on the H+ website at the following links, upon requesting a logging and password at http://hplus.ore.fr/en/database/acces-database: http://hplus.ore.fr/documents/requests/ploemeur/b3_FO_201509.csv.tgz(FO-DTS), http://hplus.ore.fr/documents/requests/ploemeur/b3_flowrate_201509.csv.tgz (flow rate), http://hplus.ore.fr/documents/requests/ploemeur/b3_Tinj_201509.csv.tgz (injection temperature) and http://hplus.ore.fr/documents/requests/ploemeur/b3_pressure_201509.csv.tgz (pressure). We thank Marie-Françoise Gerard and Thierry Labasque from the University of Rennes, Adriana Fulano from the University national of Bogota and Florian Koch from the BRGM for their precious help on the field. The authors wish to thank warmly Alain Dassargues and Serge Brouyère from the University of Liège for generously lending the portable water heater. We would like to express our gratitude to René Lefebvre from INRS for his kind and helpful comments and corrections. We finally thanks the reviewers for their constructive comments.

**Conflicts of Interest:** The funders had no role in the design of the study; in the collection, analyses, or interpretation of data; in the writing of the manuscript, or in the decision to publish the results.

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
