# Peer review of "Dipole and Convergent Single-Well Thermal Tracer Tests for Characterizing the Effect of Flow Configuration on Thermal Recovery"

_geosciences, doi:10.3390/geosciences9100440_

Round 1

Reviewer 1 Report

Authors extend the thematic issue on thermal recovery through dipole and convergent single well thermal tracer tests.

The analysis is interesting, but I have the following concerns:

1) So many related works have been listed in introduction part. Compared to the existing methods and results, what is the main contribution of this paper?

2) Authors claimed that recovery is mainly controlled by advection in convergent case. Whereas in the perfect dipole tracer test, heat exchanges by conduction within the matrix seem more significant. Add more quantitative analysis of this.

Reviewer 2 Report

General Comments

1) This is an interesting and well-written paper on an important study area. The recognition that current experiments using a thermal tracer are not “well achieved” is correct. The present study provides a detailed analysis of two types of experiments conducted in the same single borehole. As the experiments however were conducted in a single borehole and depend heavily on the interpretation of the signal from the fiber optic system, the geometry of the fracture pathways becomes very important and remains an unknown. Notably, this is the reason that similarly configured experiments have not been conducted using solute tracers, at least as evidenced by the paucity of such in the literature. When discussing “dipole” experiments, many analogous solute experiments appear in the literature but all between wells not within a single well. I believe the authors should be more forthcoming about this limitation to their results in the Abstract, Introduction (lines 72-74 should also state the disadvantage) and particularly in the Discussion and Conclusions.

2) I also object to the use of the term “dipole” or “perfect dipole” as this implies the same physical properties at each pole of the dipole, which is clearly not the case in this setting. Only a single fracture arguably intersects the injection zone, yet many fractures are exposed to the withdrawal conditions. If one were to hypothesize flow lines through a fracture network they would in no way look like the concept of a dipole in the original connotation of the term.

3) The manuscript would benefit from at least a 2-D conceptual model illustrating the fracture network that could have given rise to the observed responses. Reconstruction of the fracture orientations should be easily conducted using the televiewer results interpreted to highlight the features used for the experiments.

4) I also recognize that the innovation in this paper is the comparison between the impact on the heat distribution in the zone above the packers due to substantially different pumping rates. The results however are dependent on the deconvolution method to extract the heat response due to transport through the fractures. This method although used previously is based on several assumptions that could be influenced by the different pumping rates in the open borehole (particularly the assumption of steady state penetration of the heat from the injection tubing). Some thought and explanation around that would be helpful.

5) The features used for the experiments are of exceptionally high transmissivity, right at the top end of the range observed for this type of rock. Some comment should be added on how these results might transfer to more moderately permeable fractures, or at least provide an acknowledgement that the experimental conditions are anomalous in regards to T.

6) The cessation of injection while retaining pumping from above, imparts a significant transient to the flow field, and potentially changes flow pathways significantly. I believe this imparts some confusion in the way the temperature responds after the end of injection. This needs some comment in the discussion, particularly in lines 222-236.

7) An additional issue that subverts the interpretation of this experiment is the loss of heat to the borehole wall in the open well above the packers but below the pump. This obscures any heat inflows above the main inflow, and makes the difficulty in the deconvolution of the “injection tubing losses”. This is an unnecessary complication that could have been constrained by the use of a third packer at the top the B3-1A feature.

Specific Comments

1) Lines 103-107: A 4-m thick fracture zone is substantial. Is this a rubble zone associated with the fault? What does it look like in acoustic televiewer or optically? What orientation is this zone relative to the B3-2 feature?

2) Lines 110-112 and line 141: Some construction details on the straddle packer system should be added. For example, what is the packer mandrel made from? If it is steel, the thermal conductivity is at least an order of conductivity higher than the rock, and will provide a significant source of heat transfer into the water column directly above the upper packer. During the injection-only test (#IV, line 141) this should have been observed, if it is an issue. I see however this is explained on lines 219-221.

3) As suggested in the general comments above, the discussion in Section 2.3 is based on a porous media assumption which is not the case. I think the conceptual models shown in Figure 1 are very misleading and not representative of reality in any way.

4) Lines 166-168: These statements are very speculative. There could be conditions where advection-domination vs conduction-domination could be completely different. Flow rates would govern this for a specific case. Remove these lines, please.

5) Lines 237-244: I am reluctant to accept the explanation that “The hot inflow B3-1A corresponds to the fast advection of heat from B3-2 fracture and can be associated to the thermal tracer breakthrough”. If that pathway was connected to B3-2, the residual heat held in the rock adjacent to the pathway would show a much more sustained temperature at the B3-1A location. That is easy to show with a simple analytical model. This may be an artifact from some other source of advection where the rock does not conduct heat. More thought and explanation on this phenomenon (i.e. the immediate decline in T after the end of injection) is required.

6) Line 255: Error in expt number?

7) Equation 1: It is not clear that h(t) would be undefined as Qin goes to zero. This needs to be clarified as I do not see how the curve after the end of injection in Figure 7 is a continuous function.

8) Equation 1: The denominator remains constant for each expt and h(t) is only a function of the ratio of the Q. What is the value of Qout(T(t)-T0)? This completely unclear (i.e. is it Qout times T(t)?). I don’t believe (and must be convinced) that this equation properly expresses the heat recovery. If most of the heat is contained in the rock at the end of heat injection, this heat has to be pulled out of the rock by advective flow. The limiting factor in this case is the thermal conductivity of the rock and not necessarily the advective velocity in the fractures. I think the discussion around Figure 3 requires more thought and additional argument to support the interpretation given here.

Reviewer 3 Report

Dear Authors, Dear Editors,

You will find hereafter my review of the paper geosciences-592986 entitled “Dipole and convergent single-well thermal tracer tests for characterizing the effect of flow configuration on thermal recovery” from de La Bernardie et al. submitted in Geosciences in the special issue “Subsurface Thermography and the Use of Temperature in Geosciences”.

Although I find this paper very interesting, well written, with a good English (minor editing still required), and clearly organized, I have several major reservations about its acceptation in the present state.

I am not questioning the scientific content itself but rather the way it is presented and hence its novelty.

So, first, the novelty of this work is not explicit to me as the entire data set has already been published in de La Bernardie et al. 2018 in WRR (your reference [7]), except maybe experiment IV which is not the same experiment as experiment IV from [7]. There are new insights into it but the difference between the two papers must be better explained/highlighted.

Second, the paper lacks experimental details and figures to be understood as a standalone paper. It's annoying to always be referred to another article (in instance, your reference [7]) for the details. This paper, if accepted, will be published in open-access and as a consequence it should be complete per se.

Third, the introduction is an enormous shortcut into the state-of-the-art. You should rewrite it and properly discuss (I mean in an informative way, not just lumped references) the references you cite. I agree that concise papers are nice but not at the expense of misleading comprehension.

Fourth, the references list contains 32 papers, among which 13 or 14 are from the same research group as the authors. This is 40 % of the references list and there is a risk of bias in the manuscript. I am sure that there isn’t but still, the authors should propose a more complete introduction hence adding references from other research groups. For my line-by-line comments, I used a lot of references to support my argumentation. They can be a good starting point (but no obligation!). Since this manuscript is also submitted to the special issue, it could be a good idea to make a link with the paper already published in it or about to, especially (Comina et al., 2019; Ringel et al., 2019; Robert et al., 2019) which are close to your scope?

Fifth, I do not understand why you compare thermal recovery? Ok, to analyze the impact of flow rate but I mean the two tests you conducted do not “see” (sorry, my geophysical background) the same in the subsurface and of course, the signal you measure won’t be the same as well. It is probably just a question of wording but I missed the objective message of the study…

In brief, I missed the key message of your work if there is any?

Line-by-line comments:

Introduction:

I advise you to strengthen the introduction which go too fast to the scope of your study. Start maybe answering why heat is so much used as a groundwater tracer. There are several review paper on the topic that you could use to do so (Anderson, 2005; Irvine et al., 2015; Ptak et al., 2004). Also, you mention FO-DTS as an innovative tool to monitor heat in the subsurface but it is not that new and other geophysical methods exist, see below and a good paper from (Demirel et al., 2018).

In general, discuss the references you cite in an informative way. Do not write “these authors did that” but rather explain their findings.

1st paragraph (32-36) in general:

- These general sentences are not informative for the reader. There are many lumped references and none is properly discussed.

- Since you cite research done by the (broad) research group of Liège, I would like to correct some and suggest others.

[2] is indeed related to a heat tracer test but [3] is not at all (remove it from the references list). There are many other heat tracer tests published by Hermans and co-workers, especially single well tests which is the scope of your study (Hermans et al., 2019, 2012; Lesparre et al., 2019; Robert et al., 2019).

F0-DTS is not the only geophysical technique that can be used to get information on heat transport or storage in the subsurface. There is a review paper specifically dedicated on that (Hermans et al., 2014) and the above mentioned references all used geophysical techniques (including FO-DTS for some of them). Just to say that F0-DTS is not that new as you seem to say in your article.

[4] is ok but the data has been published first by (Wildemeersch et al., 2014) and used by [4] for modeling purposes. And by the way, [2] followed the experiment of (Wildemeersch et al., 2014) with geophysical measurements. These three papers are related to the same experiment.

37-38: … the relative low permeability of fractured rocks in general

I am not sure that you can write that like this. From my experience especially with fractured limestone (one of the biggest drinking water reservoir worldwide), it is fifty-fifty (either low or really high).

48-49: Both configurations are already used for different applications.

That's an understatement. No need to write that.

53-56: On the one hand, convergent thermal tracer tests should enhance streamline convergence to the outlet. On the other hand, perfect dipole flow tests spread heat over a larger volume of rock, enhancing heat exchanges with the rock matrix.

So basically, they do not inform on the same volume/process. It is like push/pull tests in general (Haggerty et al., 1998; Istok, 2013). If you change one parameter of the experiment (e.g. the duration of the resting phase), you change the affected volume and the final response.

I do not understand why you want to compare the energy recovery here. What is the goal behind that? (De Schepper et al., 2019) did this too but the goal was clear (to assess short-term aquifer thermal energy storage for energy flexibility purposes).

57: … the timing of energy recovery may be quite different depending on the flow configuration.

Give examples. Be informative for the reader.

61-62: In this study, we present several thermal tracer tests with continuous injection to analyze the impact of flow rate on thermal recovery.

Again, why?

66: Define FO-DTS (1st occurrence of the acronym and not defined before).

Also, you state that it is an innovative instrumentation to achieve and interpret thermal tracer tests.

I agree that they are powerful but it is no longer innovative?

See this review paper (Hermans et al., 2014) for example or this work of a former colleague (Dumont et al., 2018) that shows the same and look also at the literature related to thermal response tests (TRT) used to design closed-loop geothermal systems (BTES) (e.g. Jasmin Raymond and co-workers).

70: You justify that single-well tracer tests can be an alternative to cross-borehole ones with only [7] that presents the data of this study? It is bold.

There is a huge literature on single-well tracer tests (see the review papers mentioned above), including push-pull ones e.g. (Haggerty et al., 2001, 1998; Istok, 2013). You can certainly find some papers saying the same.

77: The setup used is fully described in [7] and summarized below.

First, present a figure with the setup and second, describe completely the setup.

The paper must be understandable per se.

82: about the thermal losses (heat losses or thermal losses but not heat thermal losses).

I anticipate other occurrences on this topic here.

You put rubber sleeves on the injection tube, what is the thermal resistance of these? I am wondering if they are really effective in this context.

About the thermal bridges you have at the junctions of the sleeves, maybe using insulating tape (eventually maintained with a suitable adhesive) (as for the thermal insulation of houses) would have helped?

Btw, you use the word/verb “isolation/(to) isolate” but in this context, you have to use “(to) insulate/insulation”. Several occurrences of this. Do a CTRL-H to change them all.

And finally, this problem mostly occurs because the injection is continuous. I am just wondering if a push-pull test could give the same (and same amount of) information about the physics you study? Because, in this case, you get rid of the thermal losses during the resting and pulling phases of the test and strongly improve the signal-to-noise ratio of your data set.

Another recommendation (maybe you can talk about that in the perspectives of you work) would be to directly install a fiber optic in new drillings (before equipping them in wells or piezometers). This means that the fiber optic is not in direct contact with the injection tube (the groundwater, the PVC or metallic casing, and the gravel pack are in between).

96: (20,000 habitant), plural and use inhabitants instead of habitants

99: plural again for meter (twice)

103-107: Again, present at least one figure. You have them from [7] and they are very useful to understand or seeing better what you write.

109-110: This phrase should come after a proper description of the setup. Here, you directly say that the setup is not presented in this paper.

Concerning this paragraph in general: it would be so much easier to understand it with a figure (I know I repeat myself).

114: You mention the boiler. It is precisely a mobile water flow heater (Swingtec AquaMobil DH7) (be precise for the reader).

130: all the night and in Table 1: 13 to 17 hours.

Would you recommend to extend the recovery duration?

The tail of the breakthrough curve contains information regarding heat transfer for the matrix / rock to groundwater. This is a good information, especially for aquifer or borehole thermal energy storage system.

Table 1: Column one, third raw: it is heated water injection rate instead of heat injection rate (or you need to change the unit).

Table 1: Test III (perfect dipole). You mention before that these tests are done with the same rate for injection and recovery (pumping). But here it’s 16 vs. 22 l/min (+33 %, +360 l in one hour). Can you comment on that? It is maybe not problematic as for tests I and II, the rate is not even in the same order of magnitude but still, it deserves a small explanation.

177-178: The same remark as for 109-110. Describe rapidly the calibration procedure.

194-196: You could test this hypothesis directly in-situ, no?

215 and 219: about 37.5 m. It would be great to add the 37.5 m limit in Fig. 2b

218: about the junctions of the sleeves and the thermal losses. Would you not recommend to insulate them? See one my remark before. These thermal bridges creates a non-negligible noise in the data. This is worth mentioning in the perspectives.

219-221: The steel tube is not insulated. Why? Not technically possible? Just asking because in hindsight it creates noise as the junctions.

Figure 2.

Do Fig. 2a and b have the same depth scale? If not, maybe add a depth scale in Fig. 2a (at least for the depths of B3-1A, B3-1B, and B3-2.

Add 37.5 m in the depth scale of Fig. 2b

Fig. 2c and Fig. 2d: Add (e.g. in top right corners of each) that it is B3-2 for c and B3-1A for d.

Fig. 2c: You should explain what happened between 6 and +-8 hours.

Section 2.4

I strongly advise to merge section 2.4 and section 3.3 as they both relates to data processing. This way, you can avoid repeating yourself in section 3.3 and ease the reading for the reader.

Minor remark concerning punctuation.

For your information, in English, at the end of a series of at least 3 items, there is a comma before the last item. Example: not a, b and c but a, b, and c. There are several occurrences in the manuscript.

255 and 257: experiment VI but in Table 1 and in the rest of the manuscript it is IV.

FO DTS or FO-DTS: Both are used in the manuscript. Select one and stick to it (FO-DTS is the most used).

273-274: In pumping configuration, the thermal losses signal is expected to be lower than without pumping due to advection of colder water along the tube.

And then in 318-319: This figure shows that temperature signal is more attenuated when the flow rate increases, which is simply due to an increase of thermal dilution with the flow rate.

You wrote the opposite interpretation of 273-274 in 318-319 (the latter being the correct one). Indeed, (related to 273-274) when pumping, it’s true that you bring back colder water along the tube. The result of this is lowering the groundwater temperature hence increasing the delta T between the T of the heated water that is injected and the T of groundwater inside the borehole. Thermal losses are directly proportional to this delta T, if it increases (the case when pumping), thermal losses increase as well.

Figure 4b (after 6 hours): The behavior is similar to Figure 2c (after 6 hours as well). I already ask in a previous comment what happened there. So same question, except that now, we know it is probably related to noise (second eigenvalue).

Figure 5: We can see that the correction you applied is basically a subtraction of 0.5 °C to the data set. Why don’t you write the result of your correction (I mean in a quantitative way) it in the manuscript?

317: … scale color … à color scale

1st paragraph of section 4 is useless, it is just a summary of what you did. Concluding is not summarizing but giving the key message and this is one of my frustration with this (overall good) manuscript: it does not provide a key message.

Last paragraph of the conclusion (line 387): why is it so important to maximize / optimize thermal recovery? You never tell it but the entire manuscript is dedicated to that.

I don’t understand. I mean if you want to study the heat exchange between groundwater in a fracture and the rock matrix, then it is best to use perfect dipole (from your results) and if you want to study heat advection, then convergent dipole are more informative. I guess this is the conclusion of your work, no?

Concerning simulations (383-384), yes it is a good idea. This is what we did in (De Schepper et al., 2019) since we could not test all possible scenarios in-situ. If you want to move forward with simulations, you can take a look at the early papers of Thomas Graf (now in Hanover) and co-workers (René Therrien e.g. in ULaval, Québec City, Canada) using HydroGeoSphere. This model can simulate everything you need.

About the key message.

From your results, we can say that convergence tests contain more information on advective processes whereas perfect dipole tests contain more info on diffusive processes (exchange between groundwater in the fracture and the rock matrix). From an experimental point of view, this is very interesting to highlight, don’t you think?

Second, your results can also give insights into heat storage applications. For example, imagine you want to preheat an aquifer to lower the delta T between the aquifer’s temperature and the temperature of the heating system in a building (the lower the delta T, the higher the coefficient of performance of the groundwater heat-pump) to offer flexibility – thermal and/or electrical see (De Schepper et al., 2019) then doing it with convergence or perfect dipole configurations won’t result in the same. Indeed, you can recover more energy (and higher T) rapidly with a convergence config but if the flexibility is not needed directly, then the perfect dipole config is better.

You should also provide the total amount of injected energy in Table 1. This also plays a role in the discussion.

392: … analogue to … analogous to? similar to? it seems to be frenglish?

Anderson, M.P., 2005. Heat as a Ground Water Tracer. Ground Water 43, 951–968. https://doi.org/10.1111/j.1745-6584.2005.00052.x

Comina, C., Giordano, N., Ghidone, G., Fischanger, F., 2019. Time-Lapse 3D Electric Tomography for Short-time Monitoring of an Experimental Heat Storage System. Geosciences 9, 167. https://doi.org/10.3390/geosciences9040167

De Schepper, G., Paulus, C., Bolly, P.-Y., Hermans, T., Lesparre, N., Robert, T., 2019. Assessment of short-term aquifer thermal energy storage for demand-side management perspectives: Experimental and numerical developments. Applied Energy 242, 534–546. https://doi.org/10.1016/j.apenergy.2019.03.103

Demirel, S., Roubinet, D., Irving, J., Voytek, E., 2018. Characterizing Near-Surface Fractured-Rock Aquifers: Insights Provided by the Numerical Analysis of Electrical Resistivity Experiments. Water 10, 1117. https://doi.org/10.3390/w10091117

Dumont, G., Robert, T., Nguyen, F., 2018. Electrical resistivity tomography and distributed temperature sensing monitoring to assess the efficiency of horizontal recirculation drains on retrofit bioreactor landfills. Geophysics 83, B13–B23. https://doi.org/10.1190/geo2016-0622.1

Haggerty, R., Fleming, S.W., Meigs, L.C., McKenna, S.A., 2001. Tracer tests in a fractured dolomite: 2. Analysis of mass transfer in single-well injection-withdrawal tests. Water Resour. Res. 37, 1129–1142. https://doi.org/10.1029/2000WR900334

Haggerty, R., Schroth, M.H., Istok, J.D., 1998. Simplified Method of “Push-Pull” Test Data Analysis for Determining In Situ Reaction Rate Coefficients. Ground Water 36, 314–324. https://doi.org/10.1111/j.1745-6584.1998.tb01097.x

Hermans, T., Lesparre, N., De Schepper, G., Robert, T., 2019. Bayesian evidential learning: a field validation using push-pull tests. Hydrogeology Journal. https://doi.org/10.1007/s10040-019-01962-9

Hermans, T., Nguyen, F., Robert, T., Revil, A., 2014. Geophysical Methods for Monitoring Temperature Changes in Shallow Low Enthalpy Geothermal Systems. Energies 7, 5083–5118. https://doi.org/10.3390/en7085083

Hermans, T., Vandenbohede, A., Lebbe, L., Nguyen, F., 2012. A shallow geothermal experiment in a sandy aquifer monitored using electric resistivity tomography. Geophysics 77, B11–B21.

Irvine, D.J., Simmons, C.T., Werner, A.D., Graf, T., 2015. Heat and Solute Tracers: How Do They Compare in Heterogeneous Aquifers? Groundwater 53, 10–20. https://doi.org/10.1111/gwat.12146

Istok, J.D., 2013. Push-Pull Tests for Site Characterization, Lecture Notes in Earth System Sciences. Springer Berlin Heidelberg, Berlin, Heidelberg. https://doi.org/10.1007/978-3-642-13920-8

Lesparre, N., Robert, T., Nguyen, F., Boyle, A., Hermans, T., 2019. 4D electrical resistivity tomography (ERT) for aquifer thermal energy storage monitoring. Geothermics 77, 368–382. https://doi.org/10.1016/j.geothermics.2018.10.011

Ptak, T., Piepenbrink, M., Martac, E., 2004. Tracer tests for the investigation of heterogeneous porous media and stochastic modelling of flow and transport—a review of some recent developments. Journal of Hydrology 294, 122–163. https://doi.org/10.1016/j.jhydrol.2004.01.020

Ringel, Somogyvári, Jalali, Bayer, 2019. Comparison of Hydraulic and Tracer Tomography for Discrete Fracture Network Inversion. Geosciences 9, 274. https://doi.org/10.3390/geosciences9060274

Robert, T., Paulus, C., Bolly, P.-Y., Koo Seen Lin, E., Hermans, T., 2019. Heat as a proxy to image dynamic processes with 4D electrical resistivity tomography. Geosciences 18.

Wildemeersch, S., Jamin, P., Orban, P., Hermans, T., Klepikova, M., Nguyen, F., Brouyère, S., Dassargues, A., 2014. Coupling heat and chemical tracer experiments for estimating heat transfer parameters in shallow alluvial aquifers. Journal of Contaminant Hydrology 169, 90–99. https://doi.org/10.1016/j.jconhyd.2014.08.001

Round 2

Reviewer 3 Report

Each of my remarks was answered.

The paper is now understandable on its own and its overall goal clearer.

I do not have additional remarks. 

Good job, nice paper!